# *AUXIN RESPONSE FACTOR 1* Acts as a Positive Regulator in the Response of Poplar to *Trichoderma asperellum* Inoculation in Overexpressing Plants

**DOI:** 10.3390/plants9020272

**Published:** 2020-02-19

**Authors:** Yue-Feng Wang, Xue-Yue Hou, Jun-Jie Deng, Zhi-Hong Yao, Man-Man Lyu, Rong-Shu Zhang

**Affiliations:** 1College of Landscape Architecture, Northeast Forestry University, 26 Hexing Road, Harbin 150040, China; wangyf777@ibcas.ac.cn (Y.-F.W.); houxueyue@nefu.edu.cn (X.-Y.H.); fireflyndmoon@nefu.edu.cn (J.-J.D.);; 2Photosynthesis Research Center, CAS Key Laboratory of Photobiology, Institute of Botany, Chinese Academy of Sciences, Beijing 100093, China; 3College of Life Sciences, Graduate University of Chinese Academy of Sciences, Beijing 100049, China

**Keywords:** poplar, *ARF1*, *Trichoderma*, growth-promotion, hormone levels, hormone signaling pathways

## Abstract

Numerous *Trichoderma* strains have been reported to be optimal biofertilizers and biocontrol agents with low production costs and environmentally friendly properties. *Trichoderma* spp. promote the growth and immunity of plants by multiple means. Interfering with the hormonal homeostasis in plants is the most critical strategy. However, the mechanisms underlying plants’ responses to *Trichoderma* remain to be further elucidated. Auxin is the most important phytohormone that regulates almost every aspect of a plant’s life, especially the trade-off between growth and defense. The AUXIN RESPONSE FACTOR (ARF) family proteins are key players in auxin signaling. We studied the responses and functions of the *PdPapARF1* gene in a hybrid poplar during its interaction with beneficial *T*. *asperellum* strains using transformed poplar plants with *PdPapARF1* overexpression (on transcription level in this study). We report that *PdPapARF1* is a positive regulator for promoting poplar growth and defense responses, as does *T*. *asperellum* inoculation. *PdPapARF1* also turned out to be a positive stimulator of adventitious root formation. Particularly, the overexpression of *PdPapARF1* induced a 32.3% increase in the height of 40-day-old poplar plants and a 258% increase in the amount of adventitious root of 3-week-old subcultured plant clones. Overexpressed *PdPapARF1* exerted its beneficial functions through modulating the hormone levels of indole acetic acid (IAA), jasmonic acid (JA), and salicylic acid (SA) in plants and activating their signaling pathways, creating similar results as inoculated with *T*. *asperellum.* Particularly, in the overexpressing poplar plants, the IAA level increased by approximately twice of the wild-type plants; and the signaling pathways of IAA, JA, and SA were drastically activated than the wild-type plants under pathogen attacks. Our report presents the potential of *ARF*s as the crucial and positive responders in plants to *Trichoderma* inducing.

## 1. Introduction

With the increasing population and environmental problems, sustainable agriculture and forestry are now in great demand. Biofertilizers and biocontrol agents who have low production costs and environmental impacts are optimal solutions for meeting such needs. Especially, *Trichoderma* spp. have received much attention for their functions as mycofungicides and plant growth promoters [1]. *Trichoderma* spp. usually exist in the rhizosphere, while some isolates can act as endosymbionts of plants [2]. The beneficial effects of *Trichoderma* are the overall outcomes of the interactions between *Trichoderma*, soil ecosystems, and plants. Accordingly, the mechanisms underlying the benefits of *Trichoderma* have been explored and uncovered from multiple perspectives [1,3,4,5,6]. One important growth-promoting mechanism is modifying the levels of phytohormones, including ethylene, cytokinin, auxin, or their related compounds in plant rhizosphere and root [3,6]. Some *Trichoderma* species were reported to produce gibberellin-related molecules (GAs) or zeatin [7]. Some can regulate the plant ethylene level by modifying the concentration of its immediate precursor, 1-aminocyclopropane-1-carboxylate (ACC) by ACC deaminase [8]. More reports demonstrated that some *Trichoderma* species could produce or degrade in vitro indole acetic acid (IAA), namely auxin, to create optimal IAA concentrations for plant growth [9,10,11].

Phytohormones regulate plant growth, development, as well as immunity and resistance against stresses through an interconnected network formed by signaling pathways [12]. Salicylic acid (SA), jasmonic acid (JA), and ethylene (ET) are crucial regulators of plant defense and resistance. Their signaling cascades cross paths with GA and IAA through hub proteins such as DELLA and EIN3 [13,14]. The biocontrol mechanisms of *Trichoderma* are highly diverse, which in turn makes *Trichoderma* spp. ubiquitously applicable agents [6]. One underlying mechanism is that *Trichoderma* activates the signaling or metabolism of SA and/or JA in plants, thus inducing systemic resistance (ISR), occasionally accompanied by systemic acquired resistance (SAR) [7,15]. Multiple reports have confirmed that *Trichoderma* inoculation can increase the levels of SA and JA, trigger ISR by SA-dependent manner while also involving JA/ET signaling pathways [16,17,18].

Under natural conditions, plants are constantly balancing between growth and defense [19]. Auxin is the key regulator of plant morphogenesis and growth [14]. In recent years, its roles as the nexus in plant-microbe interactions have emerged [20]. Auxin homeostasis in the plant is modified by concerted auxin biosynthesis, conjugation, and transport. Auxin signal transduction is achieved through binding to TRANSPORT INHIBITOR RESPONSE 1 (TIR1) and AUXIN SIGNALING F-BOX (AFB) receptors in the nucleus, which subsequently induces the proteolysis of AUXIN/INDOLE-3-ACETIC ACID (AUX/IAA) repressors and depress AUXIN RESPONSE FACTORs (ARFs) to activate the transcription of downstream auxin-responsive genes [14]. The ARF family proteins play a key role in auxin signaling and confer specificity to downstream responsive genes [21]. To date, the growth-promoting effects of *Trichoderma* spp. involving auxin signals have been attributed to the production of auxin-related compounds in vitro [9,10,11]. However, the internal responsive mechanisms in plants have not been elucidated. With such backgrounds, we studied the role of ARF1 in the interaction between a hybrid poplar *Populus davidiana* × *P. alba* var. *pyramidalis* (*PdPap*) and a beneficial *T. asperellum* strain.

## 2. Results

### 2.1. PdPapARF1 Expression Is Responsive to T. asperellum Inoculation

The DNA sequence and coding sequence (cds) of *PdPapARF1* were cloned and submitted to GenBank (with Accession No. KP165071 and KM113035.1, respectively). The coding sequence of *PdPapARF1* had 91.03%, 98.23%, and 77.80% similarities with its orthologs in *P. trichocarpa*, *Potri.004G228800.2*, *Potri.003G001000.1*, and *AtARF1* (*AT1G59750.1*) of *Arabidopsis thaliana*, respectively. In previous studies, we found that the *T. asperellum* strains ACCC32492 (Ta492) and ACCC30536 (Ta536) were both beneficial for poplar with Ta536 demonstrating the best effects among the three individual *T. asperellum* strains and that inoculation with mixed *Trichoderma* strains had even better beneficial effects [22,23]. So, we examined the expression of *PdPapARF1* in response to Ta536 or Ta492 or the combination of four *T. asperellum* strains (Ta536+Ta492+*T. asperellum* ACCC31650+*T. asperellum* T4) by quantitative real-time polymerase chain reaction (qRT-PCR). Under field conditions, *PdPapARF1* expression in the leaves and roots of one-month-old poplar plants were rapidly induced by each or the combination of *T. asperellum* strains as early as 0.5 h after inoculation (HAI). Mixed inoculation resulted in the highest expression within an early response period of 2 HAI (Figure 1).

### 2.2. Production of Transgenic Poplar with Modified Expression of PdPapARF1

To confirm that *PdPapARF1* acted as a positive regulator in the poplars inoculated with *Trichoderma*, wild type (WT) *PdPap* plants were transformed with constructed vectors to generate plants overexpressing *PdPapARF1* or with down-regulated expression of *PdPapARF1*. After screening by the expression of *PdPapARF1* through qRT-PCR, four overexpression lines (named *PdPapARF1 OX1*-4) and four RNAi lines (named *PdPapARF1* RNAi1-4) were obtained (Figure 2a). The growth traits of these transgenic poplar plants were subsequently observed.

### 2.3. PdPapARF1 Overexpression Promoted Growth and Adventitious Root Formation

Within the sterile culturing stage, transformed poplar plants with different *PdPapARF1* expression levels already demonstrated an evident difference in growth. The *PdPapARF1* OX plants showed more rapid shoot growth, increased number of adventitious roots and lateral roots compared with WT. The *PdPapARF1* RNAi plants showed severe tardiness in shoot growth, leaf size, and the number of adventitious roots (Figure 2b,c, Appendix A). In this study, our focus is on the positive effects brought by *Trichoderma* inoculation. So far, our results suggest that *PdPapARF1* is a positive responder toward *Trichoderma* inducing, and a positive regulator of poplar growth, as is *Trichoderma*. Hence, we conducted further studies only on the OX plants, using the *PdPapARF1* OX1 (referred to as OX1 in the text hereafter) line, which had the highest expression of *PdPapARF1*.

Infection with pathogen fungi may impair the growth of host plants besides causing disease symptoms. *Trichoderma*, however, is a biocontrol agent that is supposed to balance out the negative effects brought by pathogens to an extent. Hence, we treated the subcultured clones of OX1 and WT plants with sole inoculation of Ta536 on the root, or pathogenic *Alternaria alternata* (Aa) inoculation on the leaves, or duel inoculations of Ta536+Aa for 48 h, then grew the plants in soil for 40 days to monitor their growths. It takes up to thirty days for the subcultured *PdPap* plants to gradually adapt to the soil. Promoted shoot and leaf growths compared with WT were observed in the OX1 plants under each treatment at all measured time points. After 20 days, the Ta536 inoculation on WT began to show significant (*p* < 0.05) improving effects on the shoot growth (plant height). On the contrary, no promoting effects of Ta536 on the OX1 plants were observed before Day 40 (Figure 2d, Appendix A), suggesting that the overexpression of *PdPapARF1* alone was competent to promote poplar growth, namely serving the purpose of Ta536, without any extra inoculation. Our results demonstrate that *PdPapARF1* is a regulator that confers the growth-promoting benefits of *T. asperellum* inoculation. Extra Ta536 inoculation on the OX1 plants could not further improve poplar growth, suggesting other unknown mechanisms limiting the growth of poplar, possibly for maintaining the balance between vegetative growth and other life events of the plant.

### 2.4. PdPapARF1 Overexpression Altered the Response of Poplar Leaf to A. alternata Infection

Oxidative burst, which constitutes the production of reactive oxygen species (ROS) including superoxide and H_2_O_2_ at the attempted site of infection, is a rapid defend reaction of plants toward pathogen attack. ROS are essential signaling molecules when plants are under stress. However, the accumulation of ROS causes damage to plant tissues [24]. This reaction in resistant cultivars is more rapid and evident than susceptible cultivars [25]. We performed 3,3′-diaminobenzidine (DAB) staining assays to examine the reaction of OX1 plants when infected by Aa. The results reflected a much stronger oxidative burst in the OX1 plants compared with WT, indicating potentially stronger disease resistance to *A. alternata*. Furthermore, in the infected plants, the uninoculated leaves also displayed an accumulation of ROS to an extent, which was not observed in the OX1 plants (Figure 3). These results reflected ROX accumulation on a systemic scale in WT plants when subjected to biotic stress, while in the OX1 plants, the ROS accumulation was locally controlled at the infection site, giving proper protection to the uninfected plant compartments.

### 2.5. PdPapARF1 Overexpression Regulated Hormone Levels in Planta

We examined the levels of IAA, JA, and SA in the young tissues (shoot tip, ST), mature leaves (L), and roots (R) of WT and OX1 plants undergone sole Ta536 or Aa, or the duel Ta536+Aa inoculations. The overexpression of *PdPapARF1* resulted in extensive changes in the levels of these three phytohormones differentially in the three examined plant compartments. Notably, compared to WT, substantial increases of IAA, JA levels in the shoot tip, and JA, SA levels in the root were demonstrated (Figure 4, Appendix A). These findings suggest that *PdPapARF1* is a factor for regulating the levels of major growth- and defense-related hormones during poplar’s response to *Trichoderma*.

### 2.6. PdPapARF1 Overexpression Influenced the IAA, JA and SA Signaling Cascades in Planta

With the above results, subsequently, we explored whether the signaling pathways of these hormones were also influenced by *PdPapARF1* overexpression. The expression of genes corresponding to the key elements of the JA signaling cascade, *COI1*, *JAZ5*, *MYC2*, and those of the SA signaling cascade, *NPR1*, *TGA*, *PR1* were determined by qRT-PCR. Gene expression was separately investigated in three poplar compartments, namely ST, L, and R of WT, and OX1 plants undergone Ta536, Aa, or Ta536+Aa inoculations. The results demonstrated pronounced and differential changes in the expression levels of most of these genes between untreated WT and OX1 plants. When subjected to the fungal inoculation, the expression of more genes showed even more extensive changes. Notably, both JA and SA signaling pathways were drastically induced in L and R when infected with Aa (Figure 5a,b, Appendix A).

Furthermore, based on the phenotypes of OX1 plants, we examined the expression of poplar orthologs of genes that play important roles in adventitious root formation in *Arabidopsis thaliana* and poplar orthologs of auxin flux carriers including *PIN1* and *LAX3*. *AtARF6* and *AtARF8* had been confirmed as positive regulators of adventitious root initiation [26]. They controlled the initiation of adventitious root depending on the auxin signal, which inhibited the COI1 pathway of JA signaling [26,27]. Meanwhile, *AtARF6* and *AtARF8* were the positive regulators of *AtGH3.5* and *AtGH3.6*, which could regulate each other through unidentified mechanisms and regulate JA homeostasis under the depression of *AtIAA6*, *AtIAA9*, and/or *AtIAA17* [27,28]. Our results demonstrated a pronounced increase in *PdPapARF8* expression and strong positive responses of *PdPapARF1*, *PdPapARF6*, and *PdPapARF8* toward Aa inducing (Figure 5c, Appendix A). *PdPapGH3.5* and *PdPapGH3.6* showed antagonistic regulations in the OX1 plants, *PdPapGH3.6* being drastically induced, and *PdPapGH3.5* was suppressed in most cases we tested. The expression of the two poplar orthologs *PdPapIAA6-1* and *PdPapIAA6-2* were activated in the OX1 plants, both in the root and other compartments. The expression of the auxin efflux carriers *PdPapPIN1-1*, *PdPapPIN1-2*, and the auxin influx carrier *PdPapLAX3* were all largely induced in ST in the OX1 plants under all the inoculations we performed. Their expression was also largely modified in L and R under some of our treatments, especially Aa inoculation. Young shoot organs are major sites on auxin production [29]. Our results suggest that the modification of auxin fluxes is a significant event during the response of poplar to *Trichoderma* inoculation, and *PdPapARF1* is a key player in this event.

## 3. Discussion

Our results demonstrated that the overexpression of *PdPapARF1* in poplar could improve plant growth in a non-*Trichoderma*-dependent manner and had even better promoting effects than Ta536 inoculation (Figure 2d, Appendix A). The IAA, JA, SA levels, and the expression of the key genes on the signaling pathways of these hormones were altered differentially in separate poplar compartments (ST, L, and R), which possibly reflected the underlying functional mechanism of *PdPapARF1*.

The large increases of the IAA level and the expression of the auxin flux carrier genes, *PdPapPIN1*s and *PdPapLAX3*, in the young tissues (ST) reflected highly promoted auxin production and transportation in the *PdPapARF1* OX1 plants compared to WT. IAA levels in L and R of the untreated OX1 plants were also higher than WT, suggesting completely different auxin homeostasis of the OX1 plants compared to WT. These, we believe, are the reasons for the promoted growth represented by plant height and leaf count of the OX1 plants. However, when inoculated with Ta536 or Aa, the IAA levels dropped in L and R of the OX1 plants and maintained at low levels in WT (Figure 4, Appendix A). *Trichoderma* spp. might produce additional IAA-related molecules in vitro [9,10,11]. A high concentration of auxin could inhibit the growth and development of root [30]. The decrease of the IAA level in R of the OX1 plants while subjected to Ta536 inducing might be a measure of poplar to achieve optimal auxin concentration for root growth.

Most curiously, comparing the responses of the expression of the auxin-related genes examined in this study in WT or OX1 plants, multiple genes were most induced by Ta536 in the WT plants but were most induced by Aa in the OX plants instead. Genes showing such patterns included *PdPapARF1* itself, *PdPapARF8*, *PdpapPIN1-1*, *PdPapPIN1-2*, *PdPapIAA6-2* in L and R, *PdPapLAX3*, *PdPapIAA6-1* in L (Figure 5c, Appendix A). Our DAB staining assays and growth monitoring results of the WT and OX1 plants demonstrated that the OX1 plants took stronger and faster measures against the Aa invasion on-site and better secured the well-being of uninvaded plant compartments, thus promoted plant growth compared with WT (Figure 3). Hence, those genes induced by Ta536 or Aa in the opposite manners, as *PdPapARF1* itself, might be crucial factors for achieving the beneficial effects of *PdPapARF1* overexpression.

The roles as modifiers of plant hormone homeostasis, especially auxin homeostasis, of biofertilizers have been well established [31]. However, the underlying mechanisms are not yet elucidated. We believe that the modification of auxin production, distribution, fluxes, and signaling pathways in the OX1 plants compared with the WT plants were the major factors that led to the favorable growth traits of the OX1 poplars. Excessive activation of auxin-related pathways could suppress SA-dependent plant resistance but activates JA-dependent resistance in turn [32]. Pathogens can deliver cytosolic effectors to undermine the host plants’ SA-dependent immunity [33]. In the WT poplars, the Ta536 inoculation or the duel inducing of Ta536+Aa either decreased SA or JA levels in both ST and L or only induced slight increases. The SA level in ST was even down-regulated significantly by Aa infection. However, in the OX1 plants, JA levels largely increased in ST under all treatments and in L under Ta536+Aa-inducing compared to WT. The SA-dependent and JA-dependent defense mechanisms in plants are mostly antagonistic [34]. However, in the OX1 plants, the JA and SA levels in R were both elevated, and the SA production in ST was not inhibited. The SA levels in L were lower than WT but increased under the Aa invasion (Figure 4, Appendix A). The SA pathways in plants mainly respond to biotrophic pathogens, while the JA pathways respond to necrotrophic pathogens [13]. *A. alternata* is a necrotrophic fungal pathogen, which may explain the drastic increase of the JA level in the infected leaves of the OX1 plants. The distinct hormone homeostasis profiles of OX1 plants and WT plants suggest a role of *PdPapARF1* as a nexus for regulating the metabolism of major defense-related hormones, namely SA and JA.

Compared with the untreated WT plants, in the OX1 plants, the expression of JA and SA signaling-related genes was altered in the same manners as WT was by Ta536 inducing in all three examined compartments, except for a few cases, namely *PdPapMYC2* in ST, *PdPapJAZ5* in L and *PdPapTGA* in R. Such results demonstrated that the overexpression of *PdPapARF1* served highly similar purposes as Ta536 inducing concerning the SA and JA signaling pathways, although the extent of regulation varied for each gene in each poplar compartment. Notably, when infected by Aa, in the OX1 plants, the expression of most of the examined genes was up-regulated drastically in some cases. Except that *PdPapPR1* was down-regulated in ST. In contrast, in WT plants, some of these examined genes were down-regulated under Aa inducing, which was possibly due to the suppressing of plant immune responses by the pathogen [33]. Such results suggested distinct reacting mechanisms toward pathogen in the OX1 plants compared to WT. Combining our DAB staining results, the responses to pathogen in the OX1 plants might have been enhanced compared to WT. However, when Ta536 inducing was present, the examined JA and SA signaling-related genes showed varied regulation patterns in OX1 compared with WT. When Ta536 and Aa inducing were performed simultaneously, gene expression showed similar levels in OX1 and WT (Figure 5a,b, Appendix A). These results indicated that the effects of Ta536 inducing and *PdPapARF1* overexpression might have both overlaps and antagonisms on the JA and SA signaling cascades in poplar, which requests further investigations. To conclude, our study demonstrated that *PdPapARF1* was a key player in poplar’s response to Ta536, and its overexpression could benefit the poplar in similar or even better manners as did Ta536 inoculation.

## 4. Materials and Methods

### 4.1. Plant, Fungal Material, and Growth

The *Populus davidiana* × *P. alba* var. *Pyramidalis* plants were preserved by the Northeast Forestry University. Plants were subcultured on solid woody plant medium (WPM) for three weeks, then cut and cultured in liquid WPM for rooting for ten days for inoculation or transplanting into pot soil. Seedlings were cultured in a phytotron at 26 °C with a 16/8-h light/dark cycle. The *T. asperellum* strains ACCC30536, ACCC32492, and ACCC31650 were purchased from the Center of Agricultural Culture Collection of China (ACCC). The *A. alternata* strain and the *T. asperellum* strain T4 was kindly provided by Dr. Zhi-Hua Liu of Shenyang Agricultural University. Fungal strains were cultured on the potato dextrose agar (PDA) medium at 26 °C in dark for five days before being used as inoculums. Poplar plants in pot soil for monitoring growth were grown in a nursery at 26 °C with a 16/8-h light/dark cycle. Three plants were used in each treatment group as three biological replicates.

### 4.2. Cloning of PdPapARF1 Sequences and Production of Transgenic Poplars

Total RNA and genomic DNA (gDNA) were extracted from young poplar leaves using TRIzol reagent (Invitrogen, Carlsbad, CA, USA) or the DNA extraction kit of Magen (Guangzhou, China). The extracted RNA was examined with electrophoresis and NanoDrop2000 (Thermo, DE, USA). The synthesis of cDNA was performed using the RT reagent Kit with gDNA Eraser (TaKaRa, Dalian, China). The amplification of *PdPapARF1* sequences was performed using PrimeSTAR^®^ Max (TaKaRa, Dalian, China). The sequencing of amplification products was performed by Tsingke, Beijing. Sequences of *P. trichocarpa* and *A. thaliana* were acquired from Phytozome 12.

The cds of *PdPapARF1* was inserted into the pROKII expression vector at the SmaI site using the In-Fusion HD Cloning Kit (Clontech, CA, USA) to obtain the pCaMV35s::PdPapARF1 (OX) vector. Inverted repeat sequences of a 209-bp specific fragment of *PdPapARF1* were amplified and used to construct the RNAi vector pFGC-PdPap-arf1 (RNAi) (the boxed sequence in Appendix A). The designed RNAi target sequence of *PdPapARF1* was inserted into the SwaI and SmaI sites of pFGC5941 vector as inverted repeat sequences to obtain the RNAi vector. The constructed vectors were then used to obtain transgenic poplar using Agrobacterium-mediated transformation and via regeneration in callus [35]. We examined the insertion of *PdPapARF1* cds by amplifying a fragment from the gDNA using inter-extron primers to screen for overexpression plants. Plants with insertion showed both the cds amplification product and the gDNA product. A pair of primers was also used to examine the insertion of the fragment from the RNAi vector in the gDNA of the transformed plants. Meanwhile, qRT-PCR assays were performed to determine the expression of *PdPapARF1* using the TransStar Top Green qPCR SuperMix (TransGen Biotech, Beijing, China) kit and the leaves as testing material. All procedures followed the manufacturers’ instructions. Please see the supplemental material for all the primers used in this study (Appendix A).

### 4.3. Inoculation of Plants

For the data of Figure 1, plants were inoculated as described in our previous study [22,23]. For all the other data, plants were inoculated as below. The conidia of Ta536 were harvested through rinsing with 1/10 strength liquid WPM and adjusted to 10^3^ cfu/mL concentration using a hemocytometer and microscope. The adjusted conidia suspension was then used as inoculum. For Ta536 inoculation, plants were cultured with the root immersed in the inoculum for 48 h. For Aa inoculation (infection), the fifth and sixth leaves of the plant were punctured using a needle. Then, a 5-mm-diameter A. alternata mycelia disk was placed on the puncture wound, then the plant was cultured for 48 h. Both procedures were performed on one plant as duel inoculation (Ta536+Aa). Identical treatments using clean 1/10 liquid WPM and PDA disks were performed as the mock inoculation to induce the control (CK) plants. All procedures were performed under sterile conditions.

### 4.4. DAB Staining Assays, Determination of Hormone concentrations, qRT-PCR, and Statistic Analyses

Accumulation of hydrogen peroxide (H_2_O_2_) was detected using 3,3′- diaminobenzidine (DAB) staining assays, according to Hernandez et al. [36]. The two inoculated leaves (fifth, sixth) and two neighboring leaves (seventh, eighth) were stained for comparison. After the inoculation treatments (48 h), the 5-mm-length of shoot tip was harvested as ST. The inoculated leaves were harvested as L. All adventitious roots were harvested as R. The hormone concentrations were examined as described in our previous study [37]. The same plant compartments ST, L, and R were used for qRT-PCR assays as described above. All the qRT-PCR assays were performed with three internal reference genes, *PdPapACT1*, *PdPapEF1-α*, and *PdPapUBQ*, with the GenBank accession numbers KP973950, KP973951, and KP973952, respectively. Relative expression was calculated by 2^−ΔΔCt^ using the mean expression of the three internal reference genes according to a previous publication [38]. Three biological replicates of WT plants and three replicates of OX1 plant clones were performed for all the qRT-PCR assays, each with three technical replicates. Statistical analyses were performed using SigmaPlot11.0 (Systat Software Inc., San Jose, CA, USA). Data are expressed as means ± SD. One-way ANOVA and Student’s t-test following Fisher’s exact test were used to evaluate the statistical significance.

## Figures and Tables

**Figure 1 plants-09-00272-f001:**
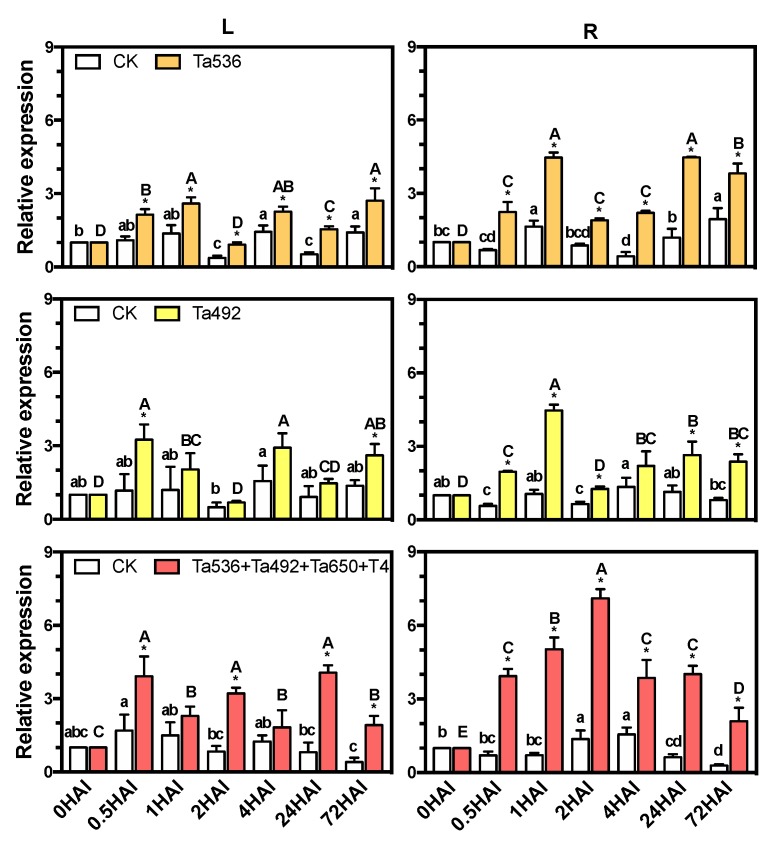
The expression of *PdPapARF1* in response to the inoculation of different beneficial *T. asperellum* strains determined by q-RT-PCR. L, mature leaf. R, root. HAI, hour(s) after inoculation. Ta650, *T. asperellum* ACCC31650. T4, *T. asperellum* T4. Different capital letters represent significant differences among the inoculated plant samples taken at different times (HAI); different lowercase letters represent significant differences among the CK plant samples taken at different times; * significant difference between inoculated and CK plants at the same time. All significances were at *p* < 0.05. All experiments were performed with three biological replicates with each replicate pooled of 10 plants and three technical replicates.

**Figure 2 plants-09-00272-f002:**
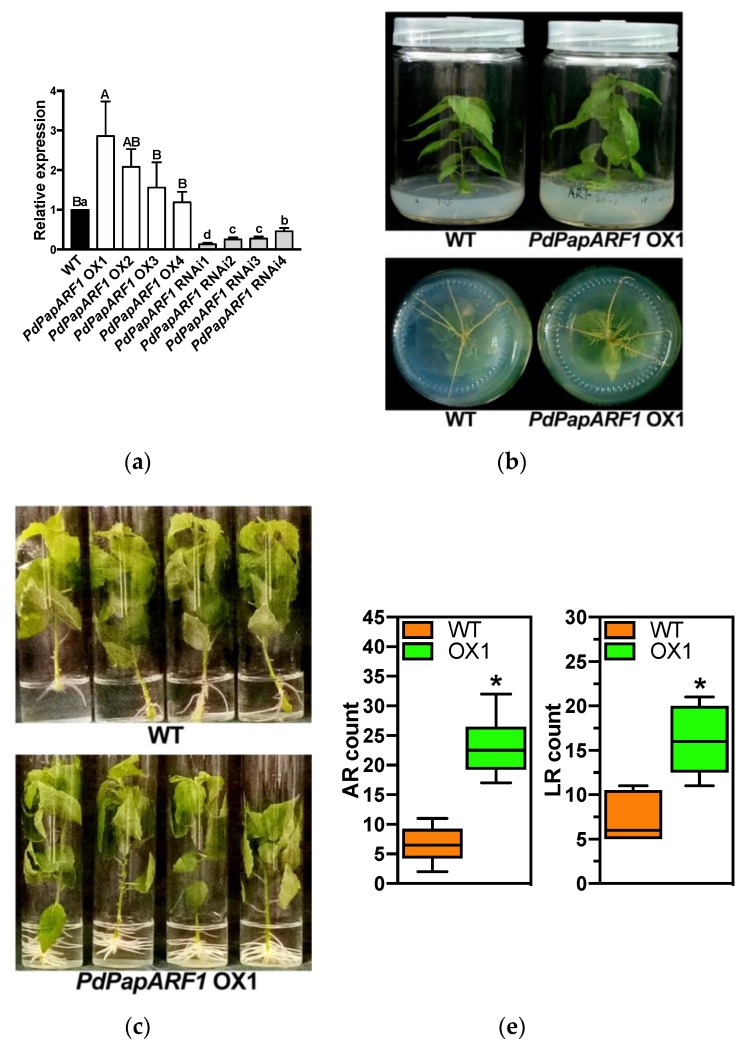
The identification, phenotypes, and growth traits of transformed poplar plants. (**a**) The relative expression of *PdPapARF1* in the transgenic plant lines determined by qRT-PCR. Data were achieved from three technical replicates. (**b**) Three-week-old subcultured poplar plants in 8 cm (bottom diameter) vessels. The two upper panels show the plants and the two lower panels show their roots. (**c**) Three-week-old poplar plants in 3.5 cm (bottom diameter) vessels showing an evident increase of adventitious roots of the OX1 plants. (**d**) Growth dynamics of the WT and OX1 plants undergone different 48-h-treatments (see methods for details), then grew in pot soil for 40 days. d, day(s). Data on 0 days did not have any significant difference between the groups; thus, those data are not shown. For each group and each treatment, n = 3. (**e**) The total amount of adventitious roots (AR) per plant (AR count, n = 8) and the total amount of lateral root (LR) per AR (LR count, n = 4). Different capital letters represent significant differences among the OX plants treated with different inoculations; different lowercase letters represent significant differences among the WT plants treated with different inoculations; * significant difference between OX and WT plants undergone the same inoculation. All significances were at *p* < 0.05.

**Figure 3 plants-09-00272-f003:**
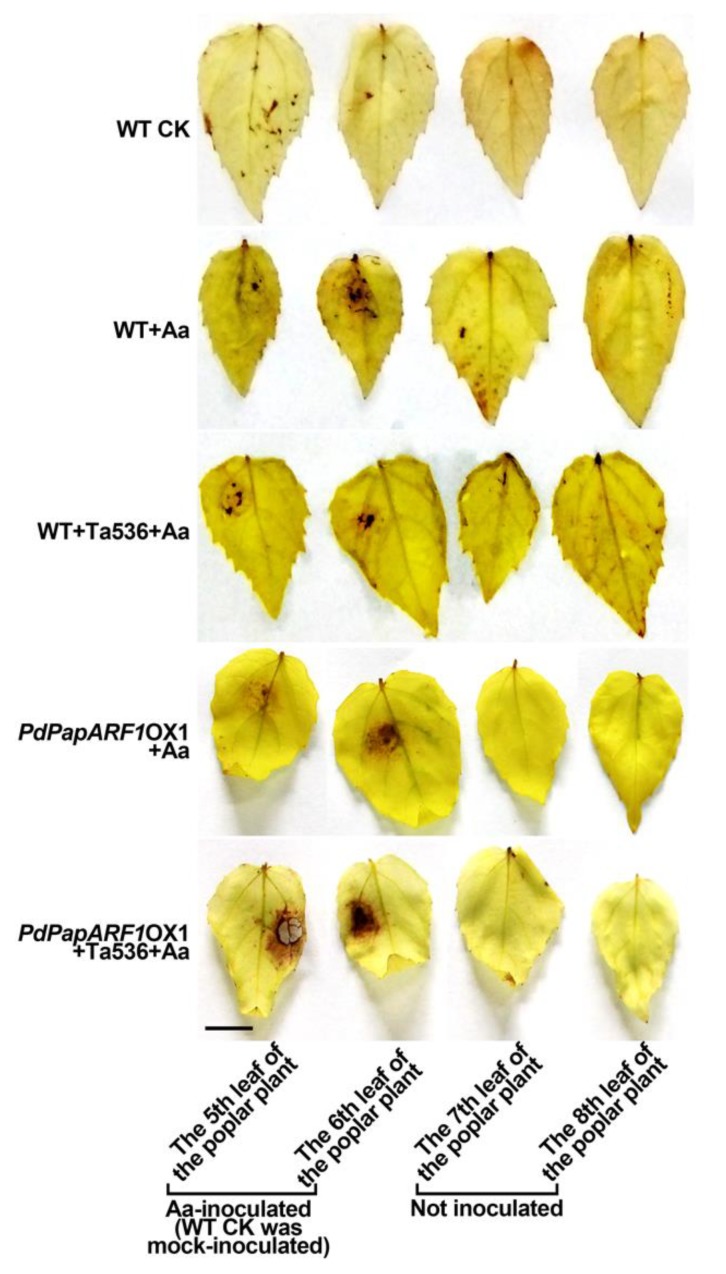
The 3,3′-diaminobenzidine (DAB) staining results of the leaves of three-week-old poplar clones subcultured under sterile conditions. Bar = 1 cm. Please note that all the inoculated leaves were punctured once, including in mock-inoculation. See details in the material and methods section.

**Figure 4 plants-09-00272-f004:**
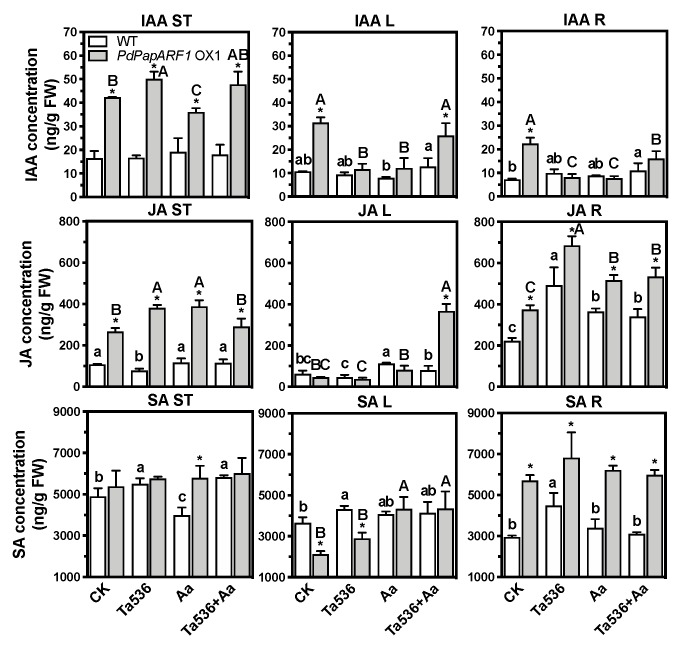
The concentrations of indole acetic acid (IAA), jasmonic acid (JA), and salicylic acid (SA). The hormone concentrations of three poplar compartments, the shoot tip (ST), inoculated leaves (L), and root (R) under no treatment (CK) or fungal treatments (Ta536, Aa, or Ta536+Aa) are shown. FW, fresh weight. Different capital letters represent significant differences among the OX plants treated with different inoculations; different lowercase letters represent significant differences among the WT plants treated with different inoculations; * significant difference between OX and WT plants undergone the same inoculation. Insignificant differences are not marked. All significances were at *p* < 0.05. All experiments were performed with three biological replicates with each replicate pooled of three plants. The replicates of OX plants were the clones of the line OX1.

**Figure 5 plants-09-00272-f005:**
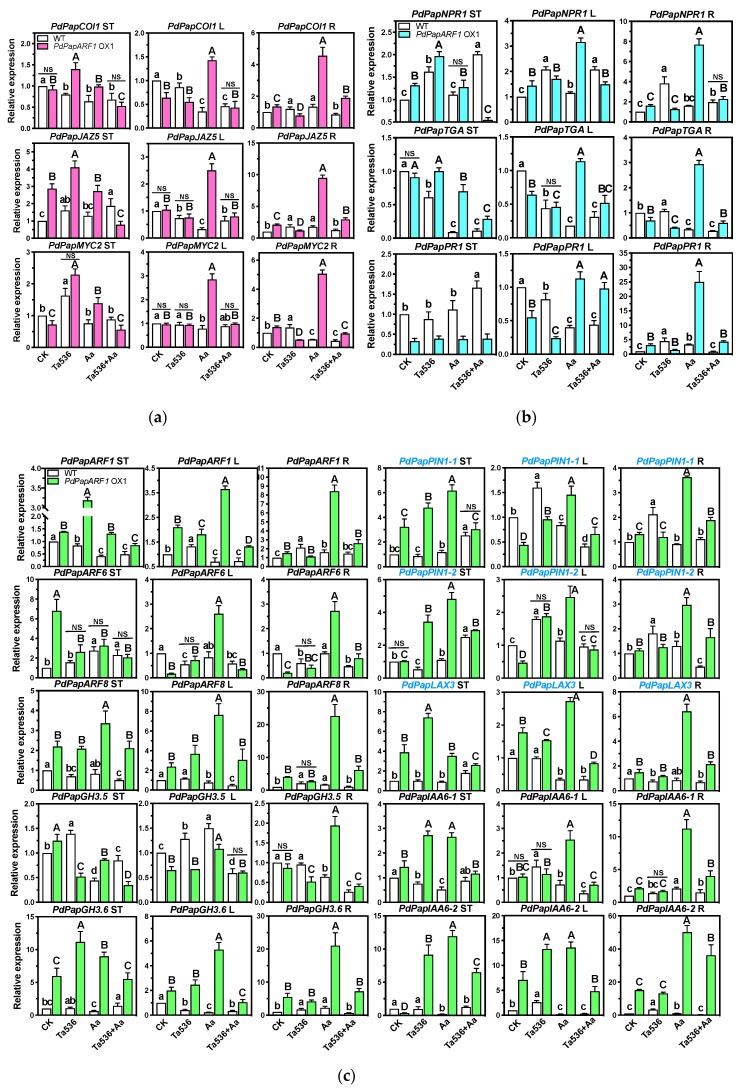
The expression of hormone-signaling-related genes of JA (**a**), SA (**b**), and IAA (**c**) in response to different fungal inoculations determined by q-RT-PCR. CK, uninoculated control. ST, shoot tip. L, inoculated leaf. R, root. Names of signaling-related genes are in black font, and names of transportation-related genes are in blue. Different capital letters represent significant differences among the OX plants treated with different inoculations; different lowercase letters represent significant differences among the WT plants treated with different inoculations; insignificant differences are not marked. NS represents that the difference between OX and WT plants undergone the same inoculation is not significant; in the results not marked with NS, significant difference exists between the OX and WT plants. All significances were at *p* < 0.05. All experiments were performed with three biological replicates with each replicate pooled of three plants and three technical replicates. The replicates of OX plants were the clones of the line OX1.

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
