# Peer review of "AUXIN RESPONSE FACTOR 1* Acts as a Positive Regulator in the Response of Poplar to *Trichoderma asperellum* Inoculation in Overexpressing Plants"

_plants, 2020, doi:10.3390/plants9020272_

Round 1
Reviewer 1 Report
The manuscript presents the role of the ARF1 gene in the growth and defense responses of hybrid poplar and studied its interaction when O.X lines inoculate with two Trichoderma strains. They showed that the overexpression of the PdPapARF1 regulates the phytohormones level (IAA, JA, and SA) through modulating the expression of the genes involved in their biosynthesis and transport and account this for better growth and development of the poplar plant. They concluded that overexpression of the PdPapARF1 had a positive effect on the growth of the poplar. Overall, the manuscript's aim is interesting, however, the designed experiments and presentation of the data is quite confusing in some cases. Further, in several cases, inaccurate labeling in the figures make it difficult to understand the reflected data and proper judgment. I believe that the current version of the manuscript must be substantially improved before any further evaluation for publication in the Plants Journal. Please refer to the comments below:
Given that ARF protein levels are regulated post-transcriptionally, the presented data here is at the mRNA level and makes it not easy to believe that these lines are overexpressed at the protein level.
In contrast to the positive effect of the Trichoderma on the plant growth, in this experiment inoculation with Ta536 and Aa did not show such effects as shown in Figures 1d and 4. The authors should explain this observation and if there is no effect from inoculation why the authors include this treatment for the subsequent experiments? What is the significance of inoculation while there is no positive effect?
The authors need to show the statistical difference in the graphs in the main text not in Supp data. Accordingly, the presence of the Table S1-S3 is not necessary as they are duplication of the data presented already in the main figures.
The abstract is too descriptive. I suggest the authors mostly reflect their obtained data in this section rather than focus on the importance of the Trichoderma and auxin hormone.
Lines96-101: The information regarding the generation of the O.X or RNAi lines need to be transferred to the method section.
Line 106: “PdPapARF1” Please use italic for gene name and plain for protein throughout the text.
P- value should be capitalized. Please correct it.
Labeling in figure 2B needs to reconsider as some lines repeated.
Figure 3 is quite confusing. “In each panel, the 8th, 7th (uninoculated), 6th and 5th (infected) leaves are shown in 145 order (left to right)”. The authors mentioned about uninoculated leaf while in the label I failed to see. The authors must include a control for this experiment and show clearly which sample is inoculated and which one not.
Lines 109-111: Please quantify the number of adventitious roots and lateral roots and put parallel to their respective images. Moreover, the authors need to include their RNAi lines inoculated with Trichoderma to and compare the results with wild type and O.X lines. In general, the authors generate the RNAi lines but did not use them in their experiments!?
Again, labeling in figure 4 is not correct regarding the SA hormone.
The discussion section needs to be improved. The authors examined several genes involved in hormone signaling in different organs shoot tip, leaf, and root. The results from each organ in some cases are different. It should be discussed and explained by the authors. Further, they used term signaling the manuscript text while all of the examined genes are not signaling genes. For instance, PIN1 is involved in auxin transport. It is recommended to categorize the genes based on their accurate function.
The manuscript needs to be revised by a native speaker.
Reviewer 2 Report
The manuscript "AUXIN RESPONSE FACTOR 1 Acts as a Positive Regulator in the Response of poplar to Trichoderma asperellum Inoculation" represents a comprehensive study on the role of ARF1 in poplar development and its reaction to Trichoderma or pathogenic fungi.
The range of provided data is considerable and the study is a substantial progress in elucidation of ARF functions. Authors demonstrated positive impact of overexpressed ARF1 gene (here PdPapARF1) on hybrid polar development and reaction to fungi. Moreover, they identified number of genes, which expression change in plants overexpressing ARF1, also during interaction with Trichoderma or Alternaria. The most important finding is identification the role of PdPapARF1 gene as a key junction in auxin-regulated plant development and pathogenesis-related response dependent on salicylic or jasmonic acid. The study is an considerable step towards deciphering of gene expression network involved in plant development and defence. In my opinion Authors could even try to outline and visualize a part of it, i.e. that regarding studied genes.
However, despite the generally positive above opinion, the paper requires major revision before possible publication. There are several unclear issues to explain and points to adjust as listed below.
The main comment
Apart from bioinformatics analysis, a function of a gene can be explained either via its overexpression or silencing/knock-out, optimally using both methods and comparison of effects. Authors initially constructed vectors both for overexpression of PdPapARF1 and for RNAi technique, as well as obtained respective transgenic plants. However after primary evaluation of the plants, they focused their studies only on overexpressing plants (in fact the one of the highest PdPapARF1 expression), completely omitting plants with silenced PdPapARF1 and without any explanation. In my opinion, studies on phytohormone levels, gene expression etc. performed on RNAi plants probably would provide interesting data (e.g. RNAi1 of the lowest expression), maybe even equivalent to these described. Therefore, to avoid ambiguities, above mentioned resignation from research on RNAi plants should be exhaustively and convincingly explained.
Alternatively, the part of research regarding RNAi plants could be completely removed from the paper. In my opinion, this would not diminish the value of work, since the range of data is absolutely appreciable. However, in the case of RNAi part removal, the text, figures, supplementary data etc. would require thorough rearrangement, as well as the title adjusted, for instance "AUXIN RESPONSE FACTOR 1 Acts as a Positive Regulator in the Response of hybrid poplar overexpressing PdPapARF1 gene to Trichoderma asperellum Inoculation"
Major remarks
Statistical differences should be shown in the respective Figures (2d, 4 and 5), instead of the tables in supplementary materials. One may supposed that Authors intended to make readable charts, hence they moved statistical analysis to the supplementary tables. However, this way very important information is not directly available, and consequently understanding of the results is more difficult and soundness of the paper slightly diminished. I would recommend to increase size of charts and put indexes of statistical differences. By the way, these indexes in the supplementary tables are also difficult to read. Supplementary tables could contain only precise numerical data. Results, the subsection 2.1. If mixed Trichoderma strains had exerted the best effect on poplar development (previous studies), while only Ta536 was chosen in the present study? Figure 1. Here could be performed statistical analysis regarding comparison of control and Ta treatments in particular timepoints (ANOVA for repeated measures). One may also consider comparison of treatments and organs (two-way ANOVA). Results, the subsection 2.2. “Inverted repeat sequences of a 209-bp specific fragment…” – which fragment? It should be clearly described or optimally marked on the sequence as a supplementary file. Figure 2a – statistical analysis is needed. Figure 2b – why only one OX plant is shown? If several RNAi plants are presented, analogously should be done for OX plants, regardless of their phenotype, even if they only slightly differed from control. Apart from that, names of plants are repeated or different plants have the same names. Images of whole plantlets and their roots should be presented in the same order. Figure 2d. Apart from remark 1, Authors could consider to present the data not according to plant×treatment but compare plants and treatments in particular days. Figure 4. Apart from remark 1, Authors could consider statistical comparison of organs (ST, L, R) regarding levels of particular phytohormones, for instance using asterisks or other symbols. Figure 5. Apart from remark 1, Authors could consider statistical comparison of organs (ST, L, R) regarding expression levels of particular genes, for instance using asterisks or other symbols. After suggested statistical analysis, probably additional trends would be found. Hence, some adjustment of the Discussion may be required.
Minor remarks
Introduction, line 47 – Trichoderma in italic font. Introduction, line 52-54. Confusing sentence. Maybe “More reports demonstrated that some Trichoderma species can produce or degrade in vitro indole acetic acid (IAA), namely auxin, to create optimal IAA concentrations for plant growth [9-11].”. Apart from that “in vitro” in italic. Results, line 87 – Trichoderma in italic font. Results, line 97. It would more correctly “PdPapARF1 cds was introduced (…)” Results, line 99. Here (and in M&M) is written “RNAi vector”, whereas in the supplementary table 4 there are two pairs of primers used for construction, suggesting two vectors. Please clarify. Figure 2 caption, line 133. “in 40 days” – after what? Of cultivation after transfer to soil or adaptation to ex vitro conditions, or post inoculation? Please complete. Figure 3 caption, line 145. “In each panel the 8th, 7th (uninoculated), 6th and 5th (infected)…” – there are only 4 leaves in a panel. Figure 4. The charts showing SA concentration are incorrectly marked (lack of L and R) Discussion, line 248. PdPapPR1 in italic font. Materials and methods, line 264. Provide full name of WPM medium. Materials and methods, line 273. From which leaves was RNA extracted for cds preparation and subsequent cloning? Untreated of inoculated – if so, with what? Materials and methods, line 277. Typo – sequencing. Materials and methods, line 280. Please adjust and complete “The cds of PdPapARF1 was inserted into the pROKII expressing vector at the SmaI site using the In-Fusion HD Cloning Kit (Clontech, USA) to obtain … vector” Materials and methods, line 283. This sentence is confusing. Please change to “The constructed vectors were then used to obtain transgenic poplar using Agrobacterium-mediated transformation and via regeneration in callus.” Moreover, provide relevant reference or describe briefly the method. Materials and methods, line 284. There is described detection of PdPapARF1 What about RNAi constructs? See also the main comment. Materials and methods, line 286. Typo – insertion. Materials and methods, line 300. Provide full name of PDA. Materials and methods, line 306 and 311. Typos – length and calculated. Tables S1-S3 – p<0.05 should be. But see the major remark 1. Table S4 - “Constructing the OX vector”. Please unify name of vectors in M&M, the main text and in this table. The full and simplified names should be provided when the first time used and in the M&M, but also in Table S4. In the text only simplified name can be used.Author Response
Please see the attachment.

Reviewer 3 Report
The manuscript by Wang et al. focuses on characterization of the role of ARF1 protein on the downstream components of auxin signaling in poplar. Authors also focus on the effects of growth promoting effect of Trichoderma fungy and performed extensive qPCR analysis of the different plant hormone response machineries. Overall, the manuscript is written well and the experiments performed in a reasonable way.
Here are some comments for the authors:
1) In none of the Figures the statistics is being present, thus it is impossible to evaluate if the presenting phenotypes are significant or not. Checking it in the supplementary documents is extremely unhandy.
2) Authors did not specify if the graphs presented are a just one replicate or it is a pool of 3 replicates
3) Why did authors choose only one line from the OX and one line from the RNAi for the further experiments? It is always better to have multiple independent line which can confirm their observed phenotype.
4) How was the duel inoculation of the fungi performed? Aa on leaves and Ta on roots of one plant?
5) Fig. 2 – For the quantification of height and leave amount the observed subtle phenotypes are not discussed and described enough. Overall, the data for the time 0 are missing. Further the statement of authors that PdPapARF1 would confer the growth-promoting benefits of T. asperellum inoculation is not true, as there is no real difference between these two samples. Why is Aa actually inducing the number of leaves at day 10 in WT compared to the non-treated control?
6) For the leaves are not cleared enough, there is still plenty of chlorophyll present and the results are not as obvious as authors are presenting. Why there are traces of ROS production in WT untreated leaves and there are none in ARFOX1? What is the ROS production in the RNAi lines?
7) Why authors omit the usage of the RNAi lines during the qPCR experiments? These should be added as well, so that authors can perform full evaluation of the effect of ARF1 gene.
8) Why Aa is inducing all the genes authors tried? Also in the roots, where if applied to the leaves it should not trigger the response?
Reviewer 4 Report
In this research, Authors describe th effect of the overexpression of an ARF Factor in poplar.
The main findings are that this overexpression changes dramatically many pathways, in particular the JA and SA signalling, strictly correlated to the plant defense toward fungal infections. These changes in gene expression are similar to the induction caused by Thrichoderma infection. Gene-Overexpressing plants were also highly reactive in producing larger oxidative bursts and ROS amount in infected leaves.
Thus, the ARF1 gene is a key factor for inducing defence-related genes and hormones pathways.
Some questions:
why did you choose 48 hours after infections of both Tricho and Alternaria for the most tests of quantitative expression? in fig 1 ARF1 seems to be highly expressed after the first 1-2 hours after Trichoderma infection, then decrease and the again increase, in leaves as well as in roots. According your opinion, which could be the reason of these up and down in the expression of ARF 1 in plants? Do you have some data (also not shown here) reporting the expression of some gene, among the many you examined, during the first 48 hours? someone could be highly expressed in the first hours (as ARF 1) while decreasing after 24/48 hours.
Minor revisions:
1. in fig. 4 : SA figures are all three titled SA ST, while probabely the second and third figure should be SA L and SA R, respectilvely.
Round 2
Reviewer 2 Report
The manuscript "AUXIN RESPONSE FACTOR 1 Acts as a Positive Regulator in the Response of poplar to Trichoderma asperellum Inoculation" has been substantially improved. Authors addressed practically all remarks of this and other reviewers, mostly in the manuscript and some others were clearly explained. The results are presented much clearer and the main findings more directly perceived.
In my opinion, the manuscript represents a truly considerable piece of data on the role of ARF1 in poplar development and its reaction to Trichoderma or pathogenic fungi. At the same time it makes solid basis for further integrative investigations on poplar-fungi interactions.
In my opinion, the manuscript requires only some minor adjustment (see below). When corrected, it can be directly accepted for publication.
Minor points
Figure 4 – graphs related to salicylic acid in shoot tips and roots. The statistical indexes are not shown for OX1 plants, one might suppose that due to insignificant differences. Anyway please, i/ either complete this, analogously to other graphs, or ii/ add phrase to the caption “insignificant differences are not marked” – after explanation of letter indexes.
Figure 5 – significant differences between WT and OX1 plants are not shown in the graphs, although for sure they were in number of cases, as well as are mentioned in the caption.
Supplementary Figure S1 caption – typo in “The phenotypes”
Supplementary material 1 – Please add short respective caption, together with explanation what means the fragment in the frame. Even if this is explained in the main text, such ‘autonomous’ short descriptions are required.
Supplementary Table S3 and S4 – please correct names of genes. There are “Pdpap…” while should be “PdPap…”
